# Evolutionary games with environmental feedbacks

Andrew R. Tilman [1✉], Joshua B. Plotkin [1] & Erol Akçay [1]

Strategic interactions arise in all domains of life. This form of competition often plays out in dynamically changing environments. The strategies employed in a population may alter the state of the environment, which may in turn feedback to change the incentive structure of strategic interactions. Feedbacks between strategies and the environment are common in social-ecological systems, evolutionary-ecological systems, and even psychological-economic systems. Here we develop a framework of 'eco-evolutionary game theory' that enables the study of strategic and environmental dynamics with feedbacks. We consider environments governed either by intrinsic growth, decay, or tipping points. We show how the joint dynamics of strategies and the environment depend on the incentives for individuals to lead or follow behavioral changes, and on the relative speed of environmental versus strategic change. Our analysis unites dynamical phenomena that occur in settings as diverse as human decision-making, plant nutrient acquisition, and resource harvesting. We discuss implications in fields ranging from ecology to economics.

[1] Department of Biology, University of Pennsylvania, Philadelphia, PA 19104, USA. ✉email: atilman@sas.upenn.edu

In many settings an individual's payoff depends on both her own strategy, or type, as well as the strategic composition in the entire population. Such interactions arise across a range of disciplines, from micro-economics to animal behavior, and they have been analyzed using game theory[1,2]. A game-theoretic analysis of competing types typically assumes that the nature of the strategic interaction is fixed in time, or that it depends on the state of an independent, exogenous environment. Real-world systems, however, often feature bi-directional feedbacks between the environment and the incentives in strategic interactions: an individual's payoff depends not only on her actions relative to the population, but also on the state of the environment, and the state of the environment is influenced by the actions adopted by individuals in the population.

Reciprocal feedbacks between strategies and the environment play out in many complex systems with broad biological and societal relevance[3–11]. In fisheries, for example, the relative reward of a high-intensity versus a low-intensity harvesting strategy depends upon the current biomass of the fish stock; and, conversely, stock dynamics depend on the frequencies of these two harvesting strategies[12]. In ecology, species strategies and interactions determine their competitive balance while also changing the abiotic environment and, in turn, the nature of competition. For example, symbiotic nitrogen fixation in plants increases local nitrogen availability over time, altering the competitive environment[13]. Conversely, nutrient availability will change incentives for nutrient exchange in such symbioses[14]. Likewise, global climate change is caused by the strategic decisions of individuals, corporations, and nations, which have long-term environmental repercussions that will, in turn, alter the strategic landscape those parties face. Feedbacks even occur in psychology, where deliberative decision-making can beneficially shape the shared environment of social norms and institutions, but that environment then sets the stage for the success of less costly, non-deliberative decision-making[15]. Examples from these diverse fields share the common feature that feedbacks between strategies and the environment fundamentally alter dynamical predictions[16–31]. Understanding such systems requires a framework for studying game-theoretic dynamics when the nature of competitive interactions influences the environment, and conversely.

Here we develop a general framework for eco-evolutionary games. We analyze the class of two-strategy, linear games linked to a renewing or decaying resource. We derive stability criteria for all strategic/environmental equilibria, and we derive the conditions that permit cyclic dynamics, bistability, or a stable equilibrium that supports multiple co-existing types. We show that environmental feedbacks alter long-term outcomes and expand the suite of dynamical possibilities. Importantly, we can characterize the large range of possible dynamical behavior in terms of a few easily interpretable quantities: incentives to lead or follow strategic changes in the population. We find that cyclical dynamics arise in a restricted subset of game structures, and that in these regions, the dynamical behavior depends critically on the relative timescale of strategic versus environmental change.

## Results

**Eco-evolutionary games**. First we describe a general framework for modeling eco-evolutionary feedbacks, and then we analyze linear two-strategy eco-evolutionary games. We characterize the range of possible dynamical behaviors in these systems, and we apply this analysis to several case studies drawn from a range of disciplines.

Eco-evolutionary games occur when evolutionary game dynamics are environmentally coupled. First, consider a set $S$ of different strategies that can be employed in a system of interest. Depending on the system being studied, the strategies may be

defined as a set of alternative resource extraction technologies, or various nutrient acquisition strategies of plants, or different physiologies or morphologies of organisms, or different cognitive types that excel in different environments. We assume that an individual's fitness from adopting strategy $s_i \in S$ is $\pi_i(\mathbf{x}, n)$. The individual's fitness is a function of her strategy, the frequency of all $K$ strategies in the population, $\mathbf{x} = [x_1, x_2, \ldots, x_K]$, which we call the population strategy profile, and also the state of the environment, $n$. Depending upon the context, the environmental state might correspond to the concentration of a vital nutrient, the biomass of species in a community, the concentration of greenhouse gasses (or other pollutants) in the atmosphere, or the quality of social norms. Since the fitness of an individual depends on the strategies employed by other individuals, the setting we have described is game-theoretic in nature.

We study the dynamics of the frequencies of strategies with the replicator equation[32], writing the rate of change of the frequency of strategy $s_i$ as

$$\dot{x}_i = \epsilon_3 x_i(\pi_i(\mathbf{x}, n) - \phi(\mathbf{x})), \tag{1}$$

where $\phi(\mathbf{x}) = \sum_{j=1}^{K} x_j \pi_j(\mathbf{x}, n)$ is the mean fitness of the population and $\epsilon_3$ is a parameter that describes the speed of strategy dynamics. This equation implies that the frequency of a strategy increases when the fitness of those who adopt it is greater than the average fitness of the population.

We have described a game-theoretic interaction that is environmentally dependent. But we are interested in systems which, furthermore, contain environmental feedbacks. Such feedbacks arise when strategies have an impact on the environment. The impact on the environment is channeled through a function, $h(\mathbf{x}, n)$, which aggregates the influence of the current population strategy profile on changes to the environmental variable $n$. The environmental factor $n$ may also have its own intrinsic dynamic governed by $f(n)$, which describes the intrinsic rate of change of the environmental variable as a function of the current environmental state. Depending on the system of study, these intrinsic dynamics could describe food webs (if modeling a higher-dimensional environmental state), soil weathering, or earth systems processes. This results in environmental change governed by

$$\dot{n} = \epsilon_1 f(n) - \epsilon_2 h(\mathbf{x}, n), \tag{2}$$

where $\epsilon_1$ and $\epsilon_2$ determine the speeds of the intrinsic environmental dynamics (independent of strategies played) and of the extrinsic impact of strategies on the environment, respectively. In total, this describes a system where evolutionary dynamics are reciprocally linked with the environment in a dynamic eco-evolutionary game. The model is described by a system of $K$ differential equations (1 environmental equation and $K - 1$ strategy equations, since $\mathbf{x}$ lies on a simplex).

For a two-strategy game with an environmental feedback we can write the eco-evolutionary system as

$$\dot{x}_1 = \epsilon_3 x_1(1 - x_1)(\pi_1(x_1, n) - \pi_2(x_1, n)), \tag{3}$$

$$\dot{n} = \epsilon_1 f(n) - \epsilon_2 h(x_1, n), \tag{4}$$

because in this case $x_1 = 1 - x_2$. This framework features three different timescales: the timescale of intrinsic dynamics of the environment, $\epsilon_1$, the timescale of the environmental impact of the strategies currently employed in the population, $\epsilon_2$, and the timescale of strategy update dynamics (strategy evolution) in the population, $\epsilon_3$. We can normalize the first two timescales relative to the third so that we drop $\epsilon_3$, without loss of generality. This framework allows for non-linearity in the payoff structures as well as in environmental impact and intrinsic dynamics, so that

the space of models and potential dynamics is vast. We focus on models in which payoffs are linear in the state of the environment and the strategy frequencies, but we also consider non-linear payoffs (see Supplementary Note 6).

**Linear eco-evolutionary games**. In this section we describe a class of two-strategy eco-evolutionary games where the payoffs to individuals are linear in both the population strategy profile, $x$, and in the environmental state, $n$. We will show that several important models from disparate fields are instances of linear eco-evolutionary games, in which we can write the payoffs in terms of a matrix

$$\Pi(n) = (1-n)\begin{bmatrix} R_0 & S_0 \\ T_0 & P_0 \end{bmatrix} + n\begin{bmatrix} R_1 & S_1 \\ T_1 & P_1 \end{bmatrix}. \quad (5)$$

Here the state of the environment, $n$, is normalized to fall between 0 and 1, and the entries of the two matrices correspond to the payoffs of the game under rich ($n = 1$) and poor ($n = 0$) environmental states. Using $\Pi(n)$, we write the payoffs for using strategy 1 and strategy 2 as $\pi_1(x, n)$ and $\pi_2(x, n)$, respectively, where $x$ denotes the fraction of the population that plays strategy 1.

In the remainder of the results section we systematically analyze linear eco-evolutionary games coupled to environments (resources) with either renewing or decaying intrinsic dynamics. We also analyze a class of models coupled to complex environments governed by tipping points.

**Eco-evolutionary games with self-renewing and decaying resources**. Intrinsic resource dynamics can take many forms, but are broadly categorized as renewing or decaying. We consider two different environmental dynamics: (i) a renewable resource where each strategy exerts degradation (or harvesting) pressure on the resource stock, and (ii) a decaying resource that is produced as a by-product of each strategy.

We first suppose there is a resource stock, $m$, that grows logistically in the absence of consumption or harvesting, and is diminished by harvesting or consumption associated with the strategies in a game. Let $e_L$ and $e_H$ be the resource harvest effort of strategies low and high, respectively, with $e_L < e_H$. The resource dynamics are then governed by

$$\frac{dm}{dt} = rm\left(1 - \frac{m}{k}\right) - qm(e_L x + e_H(1-x)), \quad (6)$$

where $x$ is the fraction of the population playing strategy L, $r$ is the intrinsic rate of resource growth, and $k$ is the resource carrying capacity. Here $q$ is a parameter that maps resource degradation pressures (or harvesting efforts) ($e_L$, $e_H$) into the rate of reduction in the resource. We assume that environmental impact rates are restricted so that $m$ will be positive at equilibrium. This implies that $e_H \in (0, r/q)$ and $e_L \in [0, e_H)$, which spans the effort the leads to maximum sustainable yield as well as the bio-economic effort level associated with open access and the tragedy of the commons. And so, such models are well suited to address questions of environmental stewardship or over-exploitation.

The resource stock $m$ can assume any non-negative value. Regardless of its initial value, though, $m$ will eventually fall between its equilibrium values when either high- or low-effort strategies dominate the population. Therefore, we can transform $m$ to an environmental state $n$ in our framework, which is bounded between 0 and 1, using a simple linear transformation between these two extreme equilibrial values (see Supplementary Note 1).

Using the payoff matrix from Eq. 5 we can write our eco-evolutionary system as

$$\dot{x} = x(1-x)(\pi_L(x, n) - \pi_H(x, n)), \quad (7)$$

$$\dot{n} = \epsilon(r - q(e_L n + e_H(1-n)))(x - n). \quad (8)$$

In terms of our general framework (Eq. 2) we have $f(n) = -(r - q(e_L n + e_H(1-n)))n$ and $h(x, n) = -(r - q(e_L n + e_H(1-n)))x$. Here we have assumed that the timescales of intrinsic resource dynamics and resource harvesting are the same (i.e., $\epsilon_1 = \epsilon_2 = \epsilon$), so that we have only one relative timescale remaining. In other words, $\epsilon$ quantifies the rate of environmental dynamics (both intrinsic and extrinsic) compared to the rate of strategy dynamics.

We can make a similar mapping for a model with a decaying resource, such as a pollutant. Let $m$ denote the pollutant level in the environment. Each strategy played in the evolutionary game produces the pollutant as a byproduct at some rate. Let $e_L$ and $e_H$ be the emissions rates of the low emissions and high emissions strategies, respectively. Then we can model the stock of $m$ with the differential equation

$$\frac{dm}{dt} = -\alpha m + e_L x + e_H(1-x), \quad (9)$$

where $x$ is the fraction of the population that plays the strategy with low emissions (strategy L), $(1-x)$ is the fraction the plays the high emission strategy and $\alpha$ is the decay rate of the resource stock. We can again define $n$, bounded between 0 and 1, as a linear transformation of $m$ (see Supplementary Note 2), yielding dynamics governed by

$$\dot{n} = \epsilon\alpha(x - n). \quad (10)$$

Although decaying and renewing resources arise in different biological or social contexts, they both yield the same qualitative results and our analysis of dynamical outcomes applies to both cases.

**Eco-evolutionary games with environmental tipping points**. An alternative model of environmental dynamics was introduced by Weitz et al.[24], in which the environment is degraded by one strategy and enhanced by the other: $\frac{dn}{dt} = \tilde{\epsilon}n(1-n)(\theta x - (1-x))$. The last term in this equation denotes environmental enhancement by one strategy at rate $\theta$ and degradation at relative rate one. Compared to the models above with intrinsic dynamics, the environment described by Weitz et al.[24] has no intrinsic dynamic: it changes only as a direct result of the strategies in the population. We contextualize the preceding work of Weitz et al.[24] within a more general framework that includes intrinsic environmental dynamics governed by tipping points; and we show that the qualitative conclusions of Weitz et al.[24] occur as a limiting special case (see case study 4, below).

**Incentives for behavioral change in eco-evolutionary games**. The dynamics of the general linear eco-evolutionary system above, with a renewing or decaying resource, can be understood in terms of four parameter combinations. These four parameters have intuitive interpretations as incentives to change behavior:

$$\Delta_L^1 = \pi_H(1, 1) - \pi_L(1, 1) = T_1 - R_1, \quad (11)$$

$$\Delta_H^1 = \pi_H(0, 1) - \pi_L(0, 1) = P_1 - S_1, \quad (12)$$

$$\delta_L^0 = \pi_L(1, 0) - \pi_H(1, 0) = R_0 - T_0, \quad (13)$$

$$\delta_H^0 = \pi_L(0, 0) - \pi_H(0, 0) = S_0 - P_0. \quad (14)$$

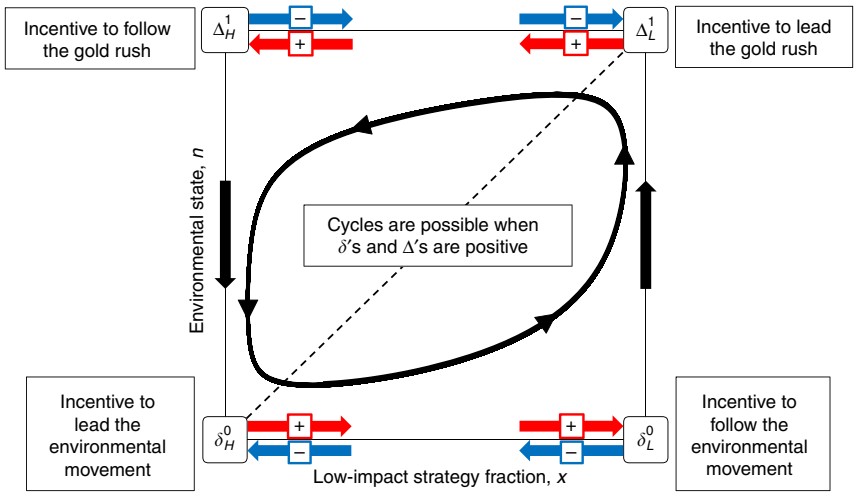

**Fig. 1 A graphical illustration of how incentive parameters in eco-evolutionary games influence dynamics.** The horizontal axis of the state space corresponds to the frequency of individuals using the strategy with low impact on the environment, whereas the vertical axis indicates the quality of the environment, $n$, with the dashed line representing the attracting environmental nullcline. Each of the four incentive parameters ($\delta$'s and $\Delta$'s) control the direction and magnitude of strategy dynamics at a corner of the state space: strategy dynamics follow the red arrows when the corresponding $\delta$ or $\Delta$ is positive, and blue arrows when negative. When all are positive, meaning there are incentives to lead and to follow strategic changes, then some form of cyclical dynamics seem plausible. However, we show that all $\delta$'s and $\Delta$'s being positive is neither necessary nor sufficient for cyclic dynamics in eco-evolutionary games.

These four parameters will allow us express dynamical outcomes in terms of the incentive to lead or follow strategy change under either a rich or a poor environmental state (Fig. 1).

The parameter $\Delta_L^1$ quantifies the incentive to switch to the strategy with high environmental impact (denoted by $\Delta$) given that all other individuals follow the low-impact strategy (denoted by the L subscript) and the system is currently in a rich environmental state (denoted by the superscript 1). In the context of socio-ecological systems, $\Delta_L^1$ can be interpreted as the incentive to "lead a gold rush"—that is, be the first player to switch to a high-impact strategy and reap the rewards of an abundant resource. By contrast, the parameter $\Delta_H^1$ quantifies the incentive to switch to the high-impact strategy under a rich environmental state, given that every other player has already switched. In other words, $\Delta_H^1$ is the incentive to "follow a gold rush".

The parameter $\delta_H^0$ quantifies incentive to switch to the low-impact strategy (denoted by $\delta$) when in a poor environmental state and when all other players are following the high-impact strategy. And so we can think of $\delta_H^0$ as the incentive to "lead an environmental movement" by reducing harvesting of a depleted resource. Finally, the parameter $\delta_L^0$ quantifies the incentive to switch to the low-impact strategy given that all other individuals are following the low-impact strategy and the environment is in a poor state. Thus $\delta_L^0$ can be seen as the incentive to "follow an environmental movement".

The verbal descriptions of these four critical parameters apply in the context of a socio-ecological system, such a fishery. But there are natural alternative interpretations of these parameters for a range of related phenomena across diverse fields.

The general linear evo-evolutionary system has up to four equilibria, by which we mean fixed points. There are two equilibria which support a single strategy in the population, at $(x^*, n^*) \in \{(0,0),(1,1)\}$; and there are also up to two equilibria that support multiple co-existing strategies in the population, denoted by $(x_+^*, n_+^*)$ and $(x_-^*, n_-^*)$. The equilibrium $(x^*, n^*) = (0,0)$ is stable only if $\delta_H^0 < 0$, which is intuitively clear from Fig. 1. Similarly, the equilibrium $(x^*, n^*) = (1,1)$ is stable only if $\Delta_L^1 < 0$.

The equilibrium at $x_+^*$ is always a saddle, and thus never stable. Whereas the equilibrium $x_-^*$ can be either stable or unstable.

We only find persistent cycles in the eco-evolutionary system when the interior equilibrium $x_-^*$ is unstable. Conditions for this equilibrium to be unstable first require

$$\Delta_H^1 + \delta_L^0 > 0 \tag{15}$$

and

$$\Delta_H^1 \delta_L^0 > \Delta_L^1 \delta_H^0. \tag{16}$$

Instability of $x_-^*$ also requires

$$\epsilon < \epsilon_{\text{crit}}, \tag{17}$$

where $\epsilon$ is the speed of environmental feedback relative to speed of strategy updating. The value of $\epsilon_{\text{crit}}$ can be expressed analytically in terms of the parameters of the system, and it differs slightly for renewing versus decaying resource feedbacks (see Supplementary Eqs. 25 and 47 in Supplementary Notes 1 and 2).

**Dynamical regimes in eco-evolutionary games**. Strategy-environment dynamics exhibit several different qualitative regimes, depending on the incentives to switch strategies ($\delta$'s and $\Delta$'s) and on the timescale separation, $\epsilon$, between strategy evolution and environmental impacts.

When there is no incentive to lead either the environmental movement or the gold rush ($\Delta_L^1, \delta_H^0 < 0$), as in Fig. 2a, then both edge equilibria are stable, and only the saddle equilibrium falls within the state space. This means that the dynamics in this regime will exhibit bistability—with attraction to a population composed entirely of one or another strategy, depending upon the initial conditions. This result is intuitive because, in this regime, there is no incentive for individuals to be leaders of change in either the poor or rich environmental state. Therefore the system will eventually be dominated by one or the other strategic type, with the corresponding environmental equilibrium.

When there are positive incentives for individuals to lead both the gold rush and the environmental movement ($\Delta_L^1, \delta_H^0 > 0$, Fig. 2b), then neither edge equilibrium is stable. In this regime,

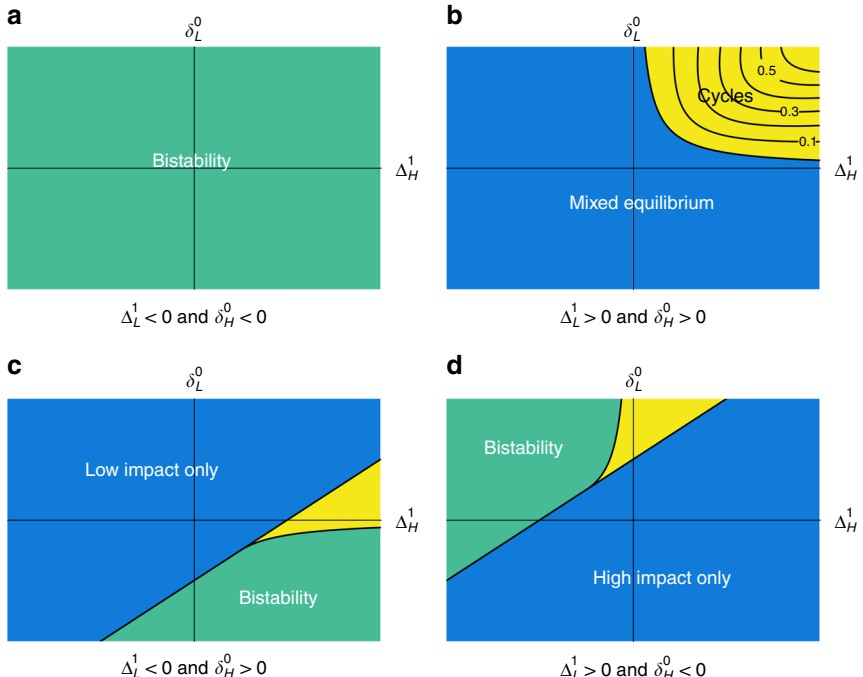

**Fig. 2 Dynamical outcomes of linear eco-evolutionary games.** Each panel shows the dynamical outcomes for different regimes of incentives to lead strategic change, $\Delta_L^1$ and $\delta_H^0$. Yellow regions denote parameter regimes that can produce limit cycles, provided $\epsilon < \epsilon_{\text{crit}}$, with level curves of $\epsilon_{\text{crit}}$ shown as black lines. Blue regions represent regions with a single dynamical outcome. Green regions represent bistability. **a** Outcomes when the incentives to lead strategic change are both negative. **b** Outcomes when incentives to lead strategic change are both positive. **c, d** Outcomes when incentives to lead strategic change are mixed. In these cases the yellow regions can exhibit bistability, dominance by one strategy, or cycles that occur in a bistable regime; the value of $\epsilon$ determines which of these outcomes occur. "Renewable resource system-level analysis" in Supplementary Note 1 provides analytical expressions for the boundaries between these dynamical regimes, in terms of the incentive parameters $\Delta_H^1, \delta_L^0, \Delta_L^1, \delta_H^0$.

because individuals are always incentivized to lead change, environmental quality can possibly cycle over time. However, positive incentives to lead change are not sufficient to induce cycles. Cycles in this regime require that the incentives to be a follower of change are also positive and stronger, in aggregate, than the incentives to lead change. Furthermore, cycles also require that the environmental feedback is sufficiently slow compared to strategy evolution (Fig. 3). In sum, when there are positive incentives to lead both movements, a stable limit cycle will occur when $\Delta_H^1 > 0$, $\delta_L^0 > 0$, $\Delta_H^1 \delta_L^0 > \Delta_L^1 \delta_H^0$, and $\epsilon < \epsilon_{\text{crit}}$. We find no evidence of cycles outside this region (see Supplementary Note 3).

When there are positive incentives to lead the environmental movement but not to lead the gold rush ($\Delta_L^1 < 0$, $\delta_H^0 > 0$; Fig. 2c), then a population composed of low-impact strategists will always be stable, whereas the high-impact strategic state will always be unstable. In this region we find parameter regimes that lead to a single monomorphic equilibrium, bistability, or stable limit cycles. In the blue-shaded region of Fig. 2c there are no interior equilibria, so the system will settle on low-impact strategists alone. In the yellow and green regions, however, there are two interior equilibria. One of these is always a saddle while the other one can be stable or unstable. In the green region, with a stable interior equilibrium the system is bistable: it will approach either a population monomorphic for low-impact strategists, or a stable mix of both strategies. In the yellow region, the stable interior equilibrium becomes unstable under slow environmental feedbacks, leading to a limit cycle or a monomorphic population depending upon the rate of environmental feedback (Fig. 4).

When there are positive incentives to lead the gold rush but not the environmental movement ($\Delta_L^1 > 0$, $\delta_H^0 < 0$; Fig. 2d), the high-impact state will always be stable and the low-impact state unstable. Here we find that, analogous to the regime in Fig. 2c,

bistability, limit cycles, and dominance by one strategy can all occur, depending on the relative incentives to follow change and the speed of environmental feedbacks.

The analysis above gives a description of the possible outcomes for a linear two-strategy eco-evolutionary system (for a detailed analysis see "Renewable resource system-level analysis" in Supplementary Note 1). The most striking result is that five easily interpretable parameters determine the qualitative properties of the system: four parameters that describe the incentives at the corners of state space, and one parameter describing the relative speed of environmental feedbacks. These parameters give immediate insight into when bistability, cyclic dynamics, mixed equilibria, or dominance by a single strategy can arise in eco-evolutionary games.

The speed of environmental feedbacks plays an important role in determining both the stability characteristics of equilibria and the basins of attraction of equilibria. Figure 4 shows the approximate basins of attraction for two equilibria under environmental dynamics of different speeds. These figures correspond to the region of potential cycles in Fig. 2c.

Our analysis in this section has assumed that payoffs are linear in the state of the environment and the frequencies of the strategies in the population. These assumptions allowed us to characterize all possible outcomes in terms of a few parameters. While these simplifying assumptions may seem to limit the range of applicability, in the remainder of the paper we highlight scientifically and societally relevant cases that fall within this model, as well as examples that extend beyond the linear framework.

**Case studies**. A large collection of prior studies on environment-strategy feedbacks, across a range of disciplines, can be be understood as linear eco-evolutionary games. The dynamical

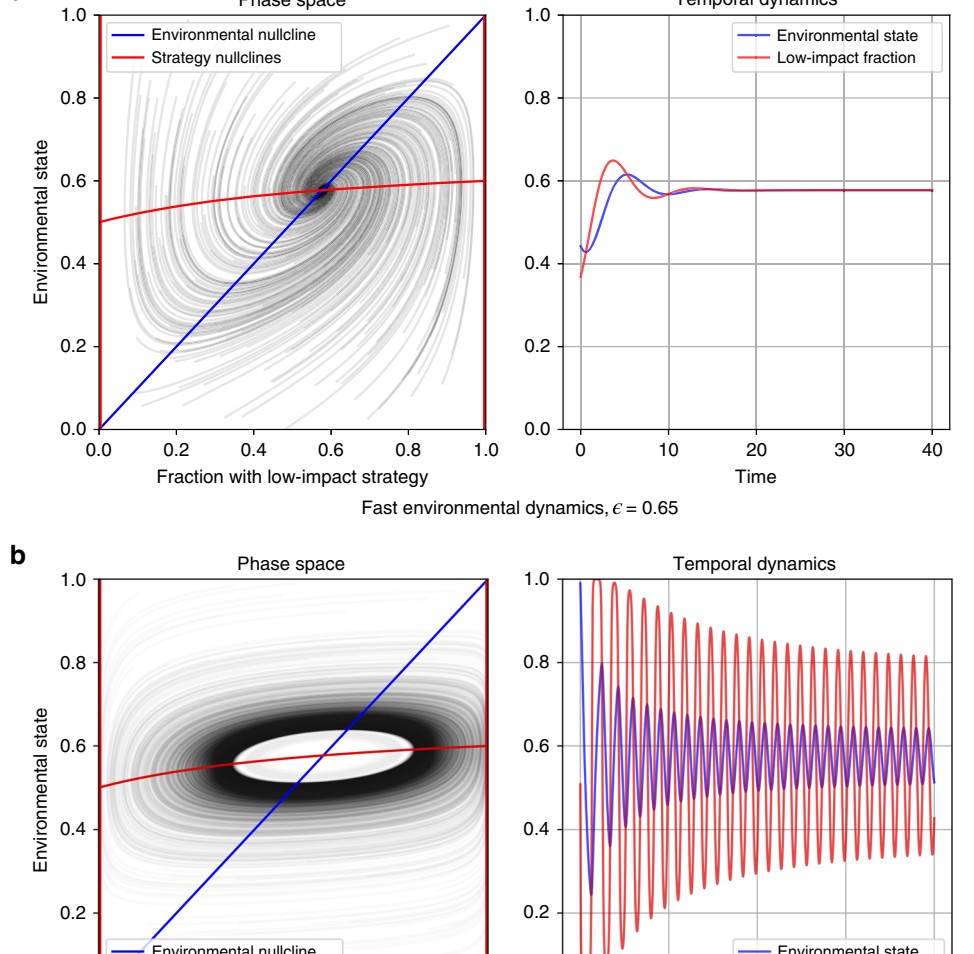

**Fig. 3 Phase planes and temporal dynamics in eco-evolutionary games with a decaying resource.** Parameters are chosen to fall in the yellow region of Fig. 2b. Only the speed of environmental dynamics relative to strategy dynamics, $\epsilon$, varies between the panels. **a** Convergence to an interior equilibrium occurs for high $\epsilon$. **b** Convergence to a limit cycle occurs for low $\epsilon$. ($\alpha = 1$, $\Delta_L^1 = 2$, $\delta_L^0 = 3$, $\Delta_H^1 = 1$, $\delta_H^0 = 1$).

properties of these models are all predicted by our analysis of these systems. In this section, we briefly review these models to highlight the broad applicability of our framework and to showcase the diversity of dynamical phenomena that occur in eco-evolutionary games.

**Case study 1: co-evolution of the environment and decision-making.** Rand et al.[15] developed a psychological model of decision-making where individuals can either make automatic hardwired decisions, or can make controlled decisions that are flexible and can shape a beneficial state of the environment. Although the motivation of their model is far from ecology, their formulation is a special case of the decaying resource model. Rand et al.[15] found that under certain circumstances these feedbacks can lead to cyclical dynamics: automatic and controlled agents cycle in abundance as the environment fluctuates in its favorability towards these the two cognitive styles. Rand et al.[15] motivate their study by noting that controlled decision making is likely to be costly but will allow individuals to choose optimal behavior. Further, they assume that when a population is dominated by controlled agents, then the fitness difference between

optimal and suboptimal decisions will decrease because institutions or public goods created by controlled agents will stabilize the environment. This then favors automated agents who choose an option quickly without paying the cost of controlled processing. Rand et al.[15] introduce a parameter that makes the cost of control frequency-dependent, so that when control agents are rare, it is more costly for them to ensure a stable environment.

The minimal model by Rand et al.[15] is a special case of our decaying resource framework, as we prove in Supplementary Note 4. By mapping their model onto our framework, the dynamical properties are immediately understood in terms of the incentive parameters of our analysis. In particular, we find that the Rand et al.[15] model falls within Fig. 2b—i.e., there are positive incentives to lead change. Embedding their model as special case of linear eco-evolutionary games allows us to show, for example, that cycles arise due to the frequency-dependent cost of being a controlled agent which assures that the model falls within the yellow region of Fig. 2b. Further, we can compute the critical time lag that produces cycles between cognitive styles, coupled with environmental cycles (see "Application to Rand et al. model" in Supplementary Note 4).

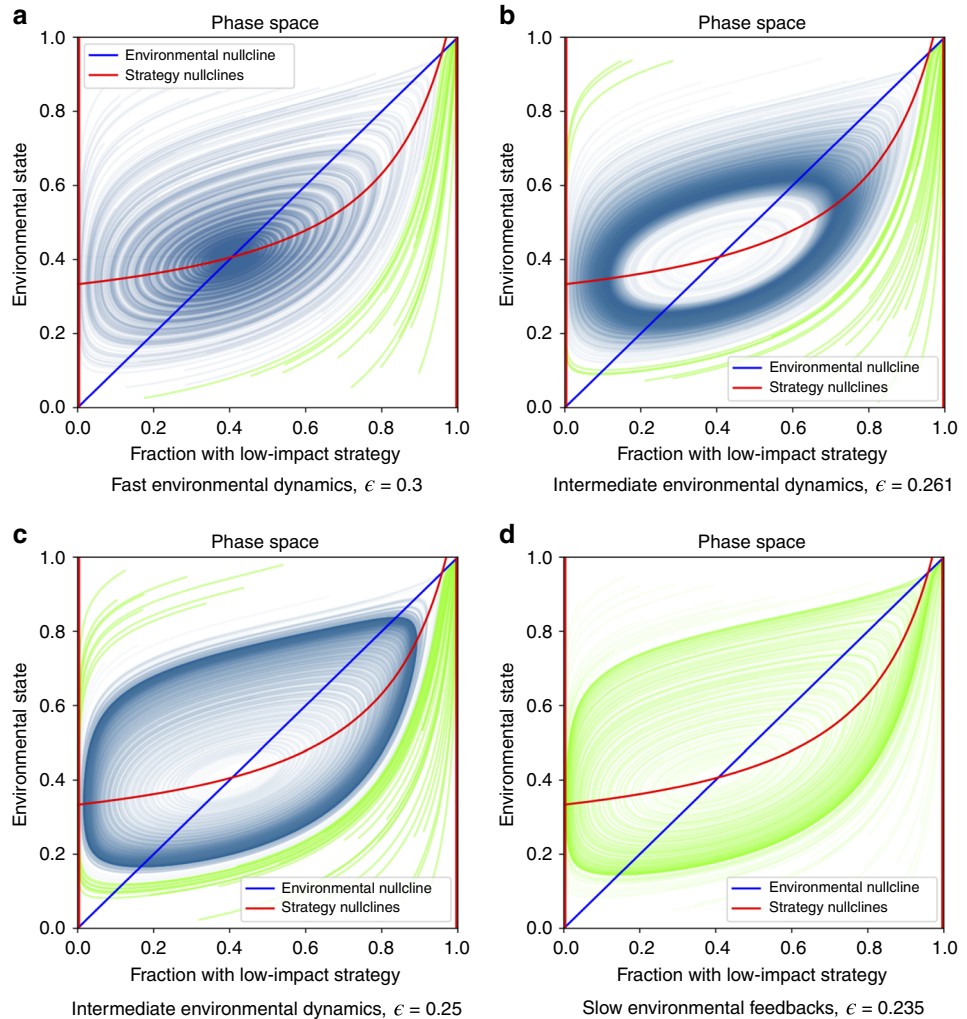

**Fig. 4 Simulations of eco-evolutionary game dynamics from the regime with possible cycles in Fig. 2c.** In each panel, dynamics proceed in a counterclockwise direction and the color of the solution curves illustrate the basins of attraction. Green curves approach the state dominated by the low-impact strategy, and blue curves represent regions that approach either the interior equilibrium or a stable limit cycle. **a** System dynamics under fast environmental feedbacks. Here, the interior equilibrium is stable and has a large basin of attraction. **b, c** Cases with environmental dynamics of intermediate speed exhibit bistability between the low-impact dominated state and limit cycles in the interior of phase space. **d** Dynamics under slow environmental feedbacks. When feedback speed falls below a critical threshold, limit cycles are no longer possible and the entire phase space approaches the low-impact equilibrium. ($\alpha = 1$, $\Delta_L^1 = -1/8$, $\delta_L^0 = 1$, $\Delta_H^1 = 4$, $\delta_H^0 = 2$).

**Case study 2: grass-legume competition**. Evo-evolutionary games provide a natural framework for studying competition between grasses and legumes. Many legumes form symbioses with nitrogen-fixing bacteria, allowing them to thrive in nitrogen-limited environments. Through time, however, some of the fixed nitrogen becomes available in the soil to nearby plants. In effect, the plant strategy of nitrogen fixation both frees plants from nitrogen limitation, and generates an environmental feedback that increases the availability of nitrogen in the soil. Grasses, on the other hand, do not fix nitrogen. Competition between obligate nitrogen-fixing legumes and grasses can be modeled as a special case of our decaying resource framework. Here the environmental state corresponds to the degree to which nitrogen is limiting, and the two strategies correspond to the species present in the system. We can thus write the relative abundance of grasses (the low nitrogen emission strategy) as $x$ and legumes (the high nitrogen emissions strategy) as $1 - x$.

We can determine the qualitative dynamics that will arise in grass-legume competition, based on the $\delta$ and $\Delta$ parameters and the relative timescale of environmental versus strategy dynamics, $\epsilon$.

First, consider the grass-dominated state ($x = 1$), with nitrogen limitation at its greatest ($n = 1$). In this state, we expect legumes to be able to invade because of the advantage of nitrogen fixation in a low-nitrogen environment. Thus we expect $\Delta_L^1 > 0$. Similarly, we expect $\delta_H^0 > 0$ because in an environment where nitrogen is not limiting ($n = 0$) that is dominated by nitrogen fixing legumes ($x = 0$), non-fixing grasses will be able to invade since they do not pay the cost of nitrogen fixation, but can reap the benefit of a nutrient-rich environment.

Therefore the dynamics will fall somewhere in Fig. 2b. Legumes are likely to have a competitive advantage in a low nitrogen environment regardless of their relative abundance, thus we expect $\Delta_H^1 > 0$. The same holds for grasses, given nitrogen is not limiting, so that $\delta_L^0 > 0$. Finally, in species competition, it is typically more difficult for the first individual to successfully invade and establish than it is for an established species to spread and increase in abundance[33]. This implies that we expect $\Delta_L^1 < \Delta_H^1$ and $\delta_H^0 < \delta_L^0$.

Because $\Delta_L^1 \delta_H^0 < \Delta_H^1 \delta_L^0$ holds we expect grass-legume systems to be susceptible to cyclic dynamics. However, cyclic dynamics

still require that the timescale of the feedback between the abundance of legumes and nitrogen availability is sufficiently slow. This too is reasonable in nature, because nitrogen is a valuable resource and a legume will tend to limit the rate at which fixed nitrogen leaks into the environment.

More broadly, feedback between plants and soil microbial communities, including through nitrogen fixation, can generate similar eco-evolutionary dynamics, with consequences for the maintenance of diversity[34,35]. A related feedback can occur between available soil nitrogen and nitrogen-fixation strategies of the rhizobium bacteria. Theoretical[36] and empirical[37] findings indicate that high nitrogen availability favors rhizobia that fix less nitrogen, while low nitrogen favors strains that fix more. This suggests that cycling may also occur in coupled nitrogen-strain frequency dynamics.

**Case study 3: common-pool resource harvesting.** Next, we consider a classic example of linked strategy and environmental dynamics: common-pool resource harvesting. There is extensive literature on common-pool harvesting, which forms the basis for bioeconomics[38]. Eco-evolutionary game theory provides a natural framework to situate common-pool resource models—because strategic interactions depend upon, and conversely influence, the abundance of the common-pool resource.

It seems plausible that even the simplest form of common-pool resource harvesting will lead to cyclic dynamics: as the biomass of resource stock collapses and overshoots, harvesters respond by reducing effort, until the resource rebounds and high-effort strategies are again profitable. Nevertheless, our analysis shows that such cycles will never occur without additional complications. We formulate common-pool resource dynamics by assuming individuals can harvest with either high, $e_H$, or low, $e_L$, effort. We let the evolutionary process on strategy frequencies be governed by a profit function,

$$\pi(e_i, \eta) = pq\eta e_i - we_i, \tag{18}$$

where $q$ is the harvest efficiency and $w$ is the marginal cost of harvest effort. This profit function maps the resource level, $\eta$, and harvest effort, $e_i$, into fitness. As in the general renewable resource model, we assume that $\eta$ is governed by logistic growth, and that the harvest rate is proportional to $\eta$ and effort ($e_L$, $e_H$). Since $\pi(e_i, \eta)$ is linear, this model is a special case of the renewable resource model we have exhaustively analyzed.

Transforming the resource level into a normalized environmental metric, we can construct a payoff matrix, $\Pi(n)$ that maps the common-pool resource harvesting model onto our framework of eco-evolutionary games. The resulting parameter values satisfy $\Delta_L^1 = \Delta_H^1 > 0$ and $\delta_H^0 = \delta_L^0 > 0$ when there are positive profits at the environmental state resulting from pure low-impact strategists and negative profits at the environment resulting from pure high-impact strategists. Under these profit assumptions, the common-pool resource system falls at the boundary of the blue and yellow regions of Fig. 2b—i.e., incentives to lead and follow change are all positive, but the there is no (positive) value of $\epsilon$ that produces cycles. And so the only possible outcome of this common-pool system is a stable mix of low-impact and high-impact strategists (see Supplementary Note 5 for detailed analysis and for other possible scenarios). However, since the system falls on the boundary of a parameter region that permits cycles, small changes to the system may induce cyclic dynamics.

We considered two extensions, introducing market pricing (where $p$ decreases as harvest quantity increases) or introducing harvesting efficiency gains (where $q$ increases as a harvest strategy becomes more common). Both extensions fall outside the scope of the linear eco-evolutionary games analyzed in this paper.

Market pricing induces non-linearity in the payoffs that harvesters receive, and frequency dependent harvest efficiency alters environmental dynamics outside of the renewing and decaying resource models considered above.

Under common-pool resource harvesting with market pricing, while the range of dynamical outcomes increases (see Supplementary Fig. 2), we do not find cyclic dynamics (see "Market pricing analysis" in Supplementary Note 6). This result occurs because market pricing effects all harvesters in the same way, and thus does not provide the extra incentive for being a follower of strategy change that can cause cycles.

Harvest efficiency may depend on strategy frequency if each strategy requires specialized skills and labor. As a strategy increases in frequency, increased opportunities for social learning may lead to increased proficiency and efficiency gains[39,40]. This effect alters both payoffs to individuals and the dynamics of the resource (see Supplementary Note 7 for analysis). As a result of these intertwined consequences, we find instances of non-monotonicity—where increasing the growth rate of the resource, for example, can first destabilize and then stabilize an interior equilibrium (see Supplementary Fig. 4). Despite this added complexity, the intuition developed from our general framework still applies. In particular, our analysis of common-pool resource harvesting as a linear eco-evolution game showed that increasing the values of either $\Delta_H^1$ or $\delta_L^0$ could cause cycles, by moving the system into the yellow region in Fig. 2b. Frequency-dependent harvest efficiency plays a similar role to increasing $\Delta_H^1$ or $\delta_L^0$, by making it more profitable to switch to a high-frequency strategy due to increased efficiency, and helps explain the cyclical dynamics that arise in this case.

**Case study 4: relationship to Weitz et al.[24].** Weitz et al.[24] developed a model of eco-evolutionary games where the environment is governed by a tipping point: it is driven to one of two extreme states in direct response to the strategies employed. Weitz et al.[24] found that the existence of persistent oscillations (specifically in the region of Fig. 2b) does not depend on the relative timescale of strategic versus environmental dynamics; whereas by contrast we find that the dynamical behavior depends critically on the relative timescale, $\epsilon$. To understand this discrepancy between models with intrinsic environmental dynamics, which are the focus of our paper, and the model of Weitz et al.[24], we analyzed a generalization of their model.

There is a simple biological motivation for environmental dynamics governed by tipping points. Imagine that the environment consists of particles (e.g., individual fish) that change state depending on the frequency of the strategies being employed. For example, individual fish might switch between being active versus hiding, depending on the feeding behavior of their predators. Weitz et al.[24] model the case when all particles have the same threshold value. More generally, one can imagine that the particles are heterogeneous, with a distribution of thresholds. Denoting the cumulative distribution function of thresholds by $F(x)$, we can write the dynamical system as

$$\dot{x} = x(1-x)(\pi_1(x, n) - \pi_2(x, n)), \tag{19}$$

$$\dot{n} = \epsilon n(1-n)(x - F^{-1}(n)), \tag{20}$$

where $n$ is the fraction of the environment that is in the state associated with strategy 1, $x$ is the frequency of strategy 1, and $F^{-1}(n)$ is the inverse of the cumulative distribution function of environmental tipping points.

For example, with a uniform distribution of tipping points centered at $\mu$ with length of $a$, the environmental dynamics are

expressed as

$$\dot{n} = \epsilon n(1-n)(x - an + a/2 - \mu). \qquad (21)$$

In the limit $a \to 0$ this model coincides precisely with the model of Weitz et al.[24] (by setting the tipping point $\mu = 1/(1 + \theta)$, and the timescale $\epsilon = (1 + \theta)\tilde{\epsilon}$), and it contains no intrinsic environmental dynamics. In this limit Weitz et al.[24] found that the existence of persistent oscillations does not depend on the relative timescales of strategic versus environmental changes, $\epsilon$.

In general, when the variability of tipping point goes to zero, Eq. 20 coincides with the model of Weitz et al.[24]. But aside from this limiting case, these environmental dynamics include an intrinsic component and, in all such cases, the existence of oscillatory solutions depends on the timescale separation between strategic and environmental change. These results hold for both the uniform distribution of tipping points (Eq. 21), as well as for distributions that produce complex non-linear dynamics. For example, under a truncated normal distribution of environmental tipping points, as the variance decreases the model again approximates Weitz et al.[24] and, despite this non-linearity, the general patterns of stability remain strikingly similar to our analysis of environments with intrinsic dynamics (Supplementary Note 8).

## Discussion
We have developed a framework to study linked environmental and evolutionary game dynamics. Our analysis provides a systematic account of dynamical outcomes for an arbitrary game with linear payoff structures, with environments that either intrinsically grow or decay. We have also analyzed examples with non-linear payoffs, or where the environment exhibits tipping points. This framework applies to any game-theoretic interaction where the strategies that individuals employ impact the environment through time, and the state of the environment conversely influences the payoffs of the game. Such feedbacks are common in social-ecological systems, evolutionary ecology, and psychology.

Despite the presence of environmental feedbacks in many systems across disciplines, analysis of feedback has previously been treated on case-by-case basis. We propose eco-evolutionary games as a synthetic framework for systematic analysis of strategic interactions with environmental feedback, allowing insight into dynamical commonalities across disparate systems. An explicit account of environmental feedbacks reveals added complexity and nuance[11]. For example, Sigdel et al.[25] have shown social norms can either cause or prevent cyclic dynamics depending on the strength of environmental feedback. By relating their model to the space of models we analyze, we can provide a synthetic understanding for why these dynamical features arise (see Supplementary Note 9). We have shown that many prior studies of strategic interactions with environmental feedbacks occur as special cases of our general framework, with an explicit mapping to the parameters of our framework. Furthermore, we have shown that the intuition developed in the simple models considered here can extend to more complex, non-linear models as well.

Perhaps the most striking result is that the rich dynamical outcomes that arise in eco-evolutionary games can be understood in terms of five intuitive parameters (see Fig. 2): the incentives to lead or follow strategy change under rich or poor environmental conditions, and the relative timescale of strategic versus environmental dynamics.

We have also generalized one of the first models of eco-evolutionary games introduced by Weitz et al.[24]. The main difference compared to our analysis is that Weitz et al.[24] studied a

feedback caused by a single environmental tipping point, in the absence of any intrinsic environmental dynamics. In many systems of ecological and social relevance the environmental variables that affect payoffs have intrinsic dynamics, meaning that they regenerate (e.g., a population) or decay (e.g., pollution) when left by themselves. Whereas Weitz et al.[24] found heteroclinic cycles independent of the timescale separation between environmental and strategy dynamics, we find that the existence of limit cycles depends critically on the degree of timescale separation, for renewing and decaying environments as well as for environments with a distribution of tipping points.

Prior work has shown that coupling strategies and the environment can induce persistent cycles even when neither the intrinsic environmental or intrinsic strategic dynamics exhibit cycles on their own[25], and our paper reaffirms this. At the same time, it is well known that many ecological systems can produce complex environmental dynamics even without feedback from individual actions[41]. Intrinsic complexity can result from multiple interacting environmental factors or a single environmental factor that is subject to age-structured or stage-structured population dynamics. Sigdel et al.[30] analyze a case where the environment has Allee effects. Allee effects can create hysteresis and critical transitions from which environmental recovery is unlikely.

Structured interactions arising from environmental heterogeneity, network structure, and spatial structure will add further complexity. Spatial models of environmental feedbacks indicate that feedbacks can lead to correlations between strategies and their environment, with consequences for the stability of cooperation[42,43]. Spatially structured interactions, and more generally, network structured interactions among individuals can fundamentally alter predictions about the strategies that will be successful in a system, but it is not well known what effects such structured interactions will have on a system where strategies and the environment are coupled. Recent work suggests that it could lead to environmental heterogeneity and the simultaneous support of many strategies with differing environmental and strategic consequences[44].

We focused most of our analysis on linear two-strategy eco-evolutionary games. And yet non-linearities in the payoff structure of strategic interactions have important effects. We have shown that introducing non-linear payoffs, through market pricing, in a common-pool resource harvesting model generates a broader class of qualitative outcomes, and the possibilities under a general non-linear payoff functions are likely even broader. Further, as the size of the strategy space increases, so does the dimensionality of the dynamical system. In these cases, a complete description of the system is not possible with only the five parameters we have considered. Nonetheless, the incentives to lead and follow strategy change may still provide valuable insights into higher-dimensional and non-linear systems. If new strategies emerge then the game itself can evolve[17], leading to novel interactions with the environment. These extensions make it clear that eco-evolutionary games can be seen as complex adaptive systems with emergent properties that are not easily predicted from the environmental dynamics or the evolutionary game structure alone.

In human societies the institutions structuring social interactions can be seen as part of the environment that co-evolve with strategic behaviors[12,45–47]. We can represent the institutional environment in our framework by constructing an institutional metric that modulates the game being played. Institutions may also interact with an explicit resource and affect the incentives individuals face in a given environment, or how strategies affect environmental dynamics. For example, the success of international environmental agreements to achieve environmental stewardship depends on how individuals and nations respond to their incentives under a changing environment. In the context of

pollution control and climate change mitigation, action is an inter-temporal public good, with benefits of individual action accruing to a large population and in the future[48]. Policies of individual nations are likely to have feedbacks on the international institutional setting, the environment, and the choices that individuals make.

Environmental feedbacks in strategic interactions are the norm, not the exception. An explicit account of these feedbacks reveals commonalities among many societally relevant systems, ranging from the psychology of decision-making to species interactions and climate-change action, and alters predictions about expected outcomes in such systems. Incorporating strategy-environment feedbacks into evolutionary game theory is paramount in future studies.

## Methods
**Analysis**. Detailed analyses supporting our key findings, illustrated in Fig. 2 and described in the section "Dynamical regimes in eco-evolutionary games", can be found in Supplementary Notes 1–3. Details of the application of our analysis to the case studies explored in the main text can be found in Supplementary Notes 4–8.

**Simulations**. Figures 3 and 4 show simulations that illustrate the importance of the relative timescale of strategic versus environmental dynamics on the long-run outcomes of eco-evolutionary games. These simulations can be replicated using the code referenced in the "Code availability" section.

**Reporting summary**. Further information on research design is available in the Nature Research Reporting Summary linked to this article.

## Data availability
All data shown in figures can be produced from the simulation code which is made freely available at https://github.com/atilman/EcoEvoGamesCode.

## Code availability
Python code used to generate figures is available at https://github.com/atilman/EcoEvoGamesCode.

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

## Author contributions

A.R.T., J.B.P., and E.A. designed the research, performed the research and wrote the paper.

## Competing interests

The authors declare no competing interests.
