## [Peer Review File · Nature Communications]

Reviewers' Comments:

Reviewer #1:

Remarks to the Author:

Summary.

The paper explores the evolutionary dynamics when there is feedback between individual strategies and the players' environment.

To this end, the authors first derive a general equation that applies to any game with K strategies and one environmental state (Section 1).

In the remainder of the paper, the authors focus on so-called linear eco-evolutionary games (meaning that individual payoffs depend linearly on the abundance of strategies and on the environmental parameter). Assuming that the environmental dynamics can be either described as a renewable or a decaying resource, the authors characterize the possible dynamics that can occur (Section 2).

In addition, the authors show that many previously suggested models (from very different areas) can all be considered as special cases of the present model (Section 3).

The SI provides all mathematical details.

General evaluation.

It was a great pleasure to read the paper. In my opinion, it is written very clearly, and it highlights interesting connections between different models from completely separate fields.

I particularly liked

- (i) the paper's focus on eco-evolutionary games,
- (ii) the intuitive interpretation of the game parameters as representing the players' incentives to lead and follow, respectively, and
- (iii) the explicit mapping from several previous models to the current framework.

Maybe the only weakness of the paper is how it deals with a previous study, the paper by Weitz et al (PNAS 2016). The two papers are clearly related: they both consider the same strategy dynamics, while they differ in how they model the environmental dynamics.

I feel that the submitted manuscript does not explain very well what previous work has accomplished, and how the present paper provides a more general (or a more useful) framework to study evolutionary games with environmental feedback.

However, once the authors better relate their model to this previous work, I don't see any reason why their paper should not be published in Nature Communications. I think the proposed model provides a very interesting perspective on eco-evolutionary games. It's great to see how many different applications can be captured by this framework (and I appreciate the explicit mapping provided in the SI).

Major comments.

(1) It is my impression that the present model has been developed completely independently of the model of Weitz et al (PNAS 2016). Indeed, the two models differ quite substantially in terms of their dynamical features (although the motivation seems to be similar). In many ways, the environmental dynamics proposed in the present model seems more realistic.

Anyway, given that the two models are clearly related, I feel the present paper needs to explain more clearly what previous work has accomplished, and why the present paper makes valuable progress. At

the moment, there is one paragraph about the paper by Weitz et al in the discussion section. I feel it would be better to discuss similarities and crucial differences more early on in the paper (for example, at the end of Section 2.1).

I would be curious to learn why the two models can yield quite different behaviors. In the discussion, the authors correctly claim that their model incorporates more realistic environmental feedbacks. Nevertheless, I find it difficult to understand on an intuitive level why the present paper finds some effects that Weitz et al didn't (for example, the critical role of epsilon for the emergence of limit cycles). After all, the eventual equations used for the environmental feedback, Eq. (6b) and (8) do not look any more complicated than the equations used by Weitz et al.

More generally, I was wondering whether the authors could represent the model of Weitz et al as a special case of the present model. To put it differently: is there some reasonable resource dynamics for the variable m , such that the transformed variable n follows the dynamics assumed in Weitz et al? I believe such a mapping would make it easier to understand the similarities and the differences between the two frameworks.

Minor comments.

(2) It seems to me that in many relevant ecological applications, there exists a tipping point — once the environment is degraded beyond a certain level, it is impossible for the environment to fully recover even if all individuals change their strategies completely.

It seems to me that the present framework cannot capture such tipping points — the environment always recovers according to some logistic growth function.

I was wondering how such tipping points could be included into the present framework (maybe by considering growth functions with a strong allee effect).

I don't ask the authors to actually do this in the present paper (the paper contains plenty of material already). However, this could be an interesting question for future research, and the authors may want to mention it in the discussion.

(3) In Figure 4, it might be useful to use some arrows to depict the direction of evolution. Also, would it be possible to explicitly show the separatrix (i.e. the boundary between the blue and the green regime)?

(4) Typos etc:

(-) Line 8: "depends not only her actions" -> "on her actions"

(-) Line 171: Instead of "equilibria", I prefer the word "fixed points" here [because there are all kinds of different equilibrium notions in game theory, but fixed points are well defined]

(-) Line 178: "we only find" -> "We only find"

(-) Lines 292 — 304. There is something wrong with the description here — at the moment, the authors provide two different explanations for why they expect $\Delta^1_H > 0$ (one in line 294, one in line 300).

(-) Eq. (12): please explicitly explain what p and q represent — at the moment this is only mentioned implicitly (the explicit explanation is only in the SI).

(-) Line 382: "And explicit account" -> "An explicit account"

(-) Line 596: "Recalling the at an interior equilibrium" -> "Recalling that at an interior equilibrium"

(-) Line 628: "will governed" -> "will be governed"

(-) Line 629: "no evidence limit cycles" -> "no evidence of limit cycles"

(-) Line 728: "We alluded the the existence" -> "We alluded to the existence"

(-) Line 731: "detial" -> "detail"

(-) Line 748: "nullcline" -> "nullcline"

(-) Line 835 "which, implies that" -> "which implies that"

Reviewer #2:

Remarks to the Author:

This paper presents a differential equation framework for modelling socio-ecological systems, shows that three study systems can be described by the framework, and uses the framework to describe some of the common features we can expect to arise from the assumptions of the framework, for instance the "gold rush" language and the finding that five parameters can describe a range of dynamical regimes. The authors use the framework to show that the dynamics of strategies and the environment depend on the incentives for individuals to lead or follow behavioural changes, and on the relative speed of environmental versus strategic change.

The paper is very well executed, with both excellent writing and figures. I found it difficult to get excited about the results, however. I think it represents a relatively small advance for the field for a number of reasons listed below, and so I find it difficult to recommend publication in a high-impact journal.

1. I am not convinced that the framework is as general as the authors claim it to be. The authors show that three study systems can be expressed in terms of the framework, but this represents a small and carefully selected cross-section of the simplest strategic systems with environmental feedback. If we had good reasons to extrapolate inductively from the three examples, this would not be a problem. However, from my experience, even modestly more structured models with environmental feedbacks can exhibit significantly more complex and highly surprising behaviour than what is described by the simplified scenarios in Figures 1-5. For example, actors having divergent perceptions of payoffs and/or various forms of population heterogeneity will in many cases produce different dynamics than the simple description in Figure 1 and expand the dimension of the parameter space beyond 5 parameters. Weird things can happen like supertransients, or surprising predictions like social pressure preventing the cycling presented in Figure 1. These rich nuances are where the frontier of much research in environmental feedback is currently happening. Because the framework only appears to capture a narrow spectrum of dynamics under environmental feedback, I do not think it has broad applicability.

2. At the same time, the kind of dynamics that are possible to capture with this framework are standard and well-known in the associated literature, despite the fact that the authors present some of those findings as significant and new. For instance, I think few researchers in the field will be surprised to find that the dynamics of strategies and the environment depend on individual incentives to lead or follow behaviour change and on the relative speed of environmental versus strategic change, especially since the latter finding is already baked into the model assumptions. Also, it is not surprising that environmental feedbacks cause the system to exhibit richer dynamics, because the dimensionality of the system increases significantly when environmental feedbacks are added (and this has also already been reported in the literature). Similarly, the cyclic dynamics report in Figure 1 are well known, even if they are not always described using the re-parameterization the authors introduce.

Overall, the authors have only provided evidence that the framework can represent the simpler types of systems with environmental feedbacks, which limits use of the framework. Also, the predicted dynamics are not novel or surprising and have been exhibited by many other models in the literature--sometimes under other rubrics like coupled human-and-natural systems, but with the same dynamics nonetheless.

Reviewer 1

Summary

The paper explores the evolutionary dynamics when there is feedback between individual strategies and the players environment. To this end, the authors first derive a general equation that applies to any game with K strategies and one environmental state (Section 1).

In the remainder of the paper, the authors focus on so-called linear eco-evolutionary games (meaning that individual payoffs depend linearly on the abundance of strategies and on the environmental parameter). Assuming that the environmental dynamics can be either described as a renewable or a decaying resource, the authors characterize the possible dynamics that can occur (Section 2).

In addition, the authors show that many previously suggested models (from very different areas) can all be considered as special cases of the present model (Section 3). The SI provides all mathematical details.

General evaluation

It was a great pleasure to read the paper. In my opinion, it is written very clearly, and it highlights interesting connections between different models from completely separate fields. I particularly liked

- (i) the papers focus on eco-evolutionary games,
- (ii) the intuitive interpretation of the game parameters as representing the players incentives to lead and follow, respectively, and
- (iii) the explicit mapping from several previous models to the current framework.

Maybe the only weakness of the paper is how it deals with a previous study, the paper by Weitz et al (PNAS 2016). The two papers are clearly related: they both consider the same strategy dynamics, while they differ in how they model the environmental dynamics. I feel that the submitted manuscript does not explain very well what previous work has accomplished, and how the present paper provides a more general (or a more useful) framework to study evolutionary games with environmental feedback.

Reply: We thank the reviewer for their encouraging comments and constructive suggestions, which have led us to undertake new research and revise the manuscript. In particular, we have worked to clarify our study in relation to Weitz et al. (2016), and, in the process, to expand the breadth of our analysis. In the revised manuscript we introduce a new class of environmental feedback relationships. This additional class of models serves to clarify how exactly our results are related to, productively extend, prior work on eco-evolutionary games. As described below, we now include a study of feedbacks arising from environmental tipping points. We show that the Weitz et al. (2016) model is a special case of this class of environmental feedbacks – and, in particular, that the particular cyclical dynamics observed by Weitz et al. (2016) occur only in the case of an environment lacking any intrinsic dynamic.

Our revised study is thus broader in its scope, while being much more precise in delineating the relationship to prior studies. We thank the referee for pushing us in this direction.

However, once the authors better relate their model to this previous work, I dont see any reason why their paper should not be published in Nature Communications. I think the proposed model provides a very interesting perspective on eco-evolutionary games. Its great to see how many

different applications can be captured by this framework (and I appreciate the explicit mapping provided in the SI).

Reply: Thanks, again, for these encouraging comments about the value of the work.

Major comments

(1) It is my impression that the present model has been developed completely independently of the model of Weitz et al (PNAS 2016). Indeed, the two models differ quite substantially in terms of their dynamical features (although the motivation seems to be similar). In many ways, the environmental dynamics proposed in the present model seems more realistic.

Anyway, given that the two models are clearly related, I feel the present paper needs to explain more clearly what previous work has accomplished, and why the present paper makes valuable progress. At the moment, there is one paragraph about the paper by Weitz et al in the discussion section. I feel it would be better to discuss similarities and crucial differences more early on in the paper (for example, at the end of Section 2.1).

Reply: The referee is correct: our conception of eco-evolutionary games was independent from Weitz et al. (2016), but strongly influenced by their work. The fundamental distinction that motivated our study was the desire to include additional realism in the environment: environmental dynamics that include some intrinsic component (such as decay or growth), unrelated to strategies being employed by organisms.

To clarify the connection to Weitz et al. (2016) the revised manuscript includes new research and writing. In section 2.1 and section 3.4 of the revised manuscript we introduce “environments with tipping points,” where the environment is composed of particles (or agents) that have intrinsic “tipping points” in the strategy frequency for changing between two states (e.g., active vs. non-active). We consider environments with heterogeneity in such tipping points. We show that this type of environmental feedback contains, as a special case, the exact model of Weitz et al. (2016), specifically when all particles have the same tipping point. Whereas when the distribution of tipping points is uniform over $[0, 1]$, the model closely resembles our linear eco-evolutionary games (the only difference being a factor of $n(1 - n)$). Analyzing models with tipping points thus allows us to directly explain the relationship between our results and those of Weitz et al. (2016). In particular, we show that the different outcomes depend on whether the environment includes any intrinsic dynamics of its own, versus being driven entirely by the agents’ strategies.

I would be curious to learn why the two models can yield quite different behaviors. In the discussion, the authors correctly claim that their model incorporates more realistic environmental feedbacks. Nevertheless, I find it difficult to understand on an intuitive level why the present paper finds some effects that Weitz et al didn’t (for example, the critical role of epsilon for the emergence of limit cycles). After all, the eventual equations used for the environmental feedback, Eq. (6b) and (8) do not look any more complicated than the equations used by Weitz et al.

Reply: The new class of (potentially non-linear) environmental feedbacks considered in the revised manuscript clarifies that the relative timescale of strategy and environmental dynamics is critical for persistent oscillations in all but the special case when the environment has no intrinsic dynamics. This is the special case that coincides with the model analyzed by Weitz et al. (2016).

More generally, I was wondering whether the authors could represent the model of Weitz et al as a special case of the present model. To put it differently: is there some reasonable resource

dynamics for the variable m , such that the transformed variable n follows the dynamics assumed in Weitz et al? I believe such a mapping would make it easier to understand the similarities and the differences between the two frameworks.

Reply: The revised manuscript now contains a broad class of models that contains Weitz et al. (2016) as a special case. Moreover, the original set of models that we analyzed are also closely related to this class and have the same dependence on ϵ for stability and oscillations.

Minor comments

(2) It seems to me that in many relevant ecological applications, there exists a tipping point once the environment is degraded beyond a certain level, it is impossible for the environment to fully recover even if all individuals change their strategies completely. It seems to me that the present framework cannot capture such tipping points the environment always recovers according to some logistic growth function. I was wondering how such tipping points could be included into the present framework (maybe by considering growth functions with a strong allee effect). I don't ask the authors to actually do this in the present paper (the paper contains plenty of material already). However, this could be an interesting question for future research, and the authors may want to mention it in the discussion.

Reply: This is indeed an interesting point, and we thank the reviewer for bringing it up. We have added an entire section on environmental dynamics governed by tipping points. And we now also discuss the possibility that referee highlights in a paragraph in the Discussion, where we also discuss environments that can lead to hysteresis and other complex dynamics, including Allee effects.

(3) In Figure 4, it might be useful to use some arrows to depict the direction of evolution. Also, would it be possible to explicitly show the separatrix (i.e. the boundary between the blue and the green regime)?

Reply: While we do not explicitly show the separatrix, the colors of the solution curves, and the regions of state space they cover makes a reasonable approximation of the basins of attraction of the two equilibria. To reduce confusion, we note in the caption that the dynamics follow counter clockwise trajectories.

Typos, etc.

Line 8: “depends not only her actions” → “on her actions”

Line 171: Instead of “equilibria”, I prefer the word “fixed points” here [because there are all kinds of different equilibrium notions in game theory, but fixed points are well defined]

Reply: We changed as suggested.

Line 178: “we only find” → “We only find”

Lines 292 - 304. There is something wrong with the description here - at the moment, the authors provide two different explanations for why they expect $\Delta_H^1 > 0$ (one in line 294, one in line 300).

Reply: This error has been corrected, thank you for bringing this to our attention.

Eq. (12): please explicitly explain what p and q represent - at the moment this is only mentioned implicitly (the explicit explanation is only in the SI).

Reply: Thank you for pointing this out, we have added descriptions of these parameters to the main text.

Line 382: “And explicit account” → “An explicit account”

Line 596: “Recalling the at an interior equilibrium” → “Recalling that at an interior equilibrium”

Line 628: “will governed” → “will be governed”

Line 629: “no evidence limit cycles” → “no evidence of limit cycles”

Line 728: “We alluded the the existence” → “We alluded to the existence”

Line 731: “detial” → “detail”

Line 748: “nullcline” → “nullecline”

Line 835 “which, implies that” → “which implies that”

Reply: We have corrected all the noted typos, and thank the reviewer for catching them.

Reviewer 2

This paper presents a differential equation framework for modelling socio-ecological systems, shows that three study systems can be described by the framework, and uses the framework to describe some of the common features we can expect to arise from the assumptions of the framework, for instance the gold rush language and the finding that five parameters can describe a range of dynamical regimes. The authors use the framework to show that the dynamics of strategies and the environment depend on the incentives for individuals to lead or follow behavioural changes, and on the relative speed of environmental versus strategic change.

Reply: We agree with the reviewer that our paper can describe features of social-ecological systems. We hasten to add that applications of our framework are not limited to socio-ecological systems. As our examples illustrate, there are many cases drawn from natural or socio-political or even psychological dynamics that can be described in our framework – such as the study of Rand et al. (2017) on controlled versus automatic cognition.

The paper is very well executed, with both excellent writing and figures. I found it difficult to get excited about the results, however. I think it represents a relatively small advance for the field for a number of reasons listed below, and so I find it difficult to recommend publication in a high-impact journal.

Reply: We respectfully disagree with the reviewer's evaluation of the advance our paper represents – and we believe the disagreement arises from poor presentation of the original manuscript, which may have erroneously suggested novelty in the dynamical outcomes observed. Our contribution is not to provide a specific model of socio-ecological systems that advances that particular field, or discovers new dynamics, but rather to provide a general analysis of a large class of models, and to show that many different cases in natural and social dynamics (not limited to, socio-ecological systems) can be mapped into this class of models. We believe that there is intrinsic value in demonstrating commonalities between dynamics in very disparate domains of inquiry; and providing a synthetic framework to analyze these dynamics in terms of a few, intuitive parameters.

Moreover, our work highlights the importance of timescale separation between environmental and strategic dynamics in shaping dynamical outcome – a result that seems to contradict a very prominent recent paper on eco-evolutionary games (for reasons we have detailed in the revised manuscript).

1. I am not convinced that the framework is as general as the authors claim it to be. The authors show that three study systems can be expressed in terms of the framework, but this represents a small and carefully selected cross-section of the simplest strategic systems with environmental feedback. If we had good reasons to extrapolate inductively from the three examples, this would not be a problem. However, from my experience, even modestly more structured models with environmental feedbacks can exhibit significantly more complex and highly surprising behaviour than what is described by the simplified scenarios in Figures 1-5.

Reply: The case studies we present were indeed selected to illustrate the breadth of possible applications of eco-evolutionary games. They were selected because they are models that capture the essential dynamics of the phenomena they each represent. It is true that linear eco-evolutionary games are undoubtedly a special case of the (very large) universe of models in this realm. Nonetheless, we believe

it valuable to demonstrate how many different problems can be mapped onto this framework, and how common insights can be drawn from their joint analysis.

In the revised manuscript we have included a whole new class of environmental feedbacks (based on tipping points) that include complex non-linear dynamics as well; and we show, in the supplement, that the stability criteria in these systems are still well characterized by the five intuitive parameters that we focus on in the main text.

For example, actors having divergent perceptions of payoffs and/or various forms of population heterogeneity will in many cases produce different dynamics than the simple description in Figure 1 and expand the dimension of the parameter space beyond 5 parameters. Weird things can happen like supertransients, or surprising predictions like social pressure preventing the cycling presented in Figure 1. These rich nuances are where the frontier of much research in environmental feedback is currently happening. Because the framework only appears to capture a narrow spectrum of dynamics under environmental feedback, I do not think it has broad applicability.

Reply: These examples of complexities in environmental feedback are indeed fascinating and we thank the reviewer for pointing us to them. We have some found excellent examples of models that contains many of these features, including several recent papers by Sigdel et al. (2017, 2019). In particular, Sigdel et al. (2017) show that increasing the strength of social norms can preclude cyclic dynamics in forest cover and conservation attitudes of the public. We have revised our study to include a detailed discussion of this work and its relationship to the models we analyze (see lines 438-441 in the main text, and supplementary information section S9).

A closer look to these examples strengthens our argument, as we find that the surprising complexities uncovered by Sigdel et al. (2017) can be understood by our analysis.

In particular, Sigdel et al. (2017) analyze the model

$$\dot{x} = \kappa x(1-x)[c - F + \xi(2x - 1)] \quad (1a)$$

$$\dot{F} = RF(1-F) - h(1-x)F \quad (1b)$$

where x is the fraction of the population with a pro-conservation opinion, F is the degree of forest cover, c is the conservation value of forest, κ is the social learning rate, ξ is the strength of social norms, R is the regeneration rate of forest, and h is the harvest rate.

Our primary approach is to consider the incentives to change strategies at the corners of the state space, where the corners are defined as the long run steady states of the environment that are approached under dominance by a single strategy. In the model of Sigdel et al. (2017) the location of these corners depends on whether $h > R$ or not.

In the case that $h < R$, we can write the corner incentives as

$$\Delta_L^1 = 1 - c - \xi$$

$$\delta_H^0 = c - 1 + h/R - \xi$$

$$\Delta_H^1 = 1 - c + \xi$$

$$\delta_L^0 = c - 1 + h/R + \xi,$$

and when $h > R$, the corner incentives are

$$\begin{aligned}\Delta_L^1 &= 1 - c - \xi \\ \Delta_H^1 &= 1 - c + \xi \\ \delta_H^0 &= c - \xi \\ \delta_L^0 &= c + \xi,\end{aligned}$$

where Δ_L^1 and δ_H^0 are the incentives to lead change of opinion under high and low forest cover, respectively. Δ_H^1 and δ_L^0 and the incentives to follow change of opinion under high and low forest cover, respectively.

Using our approach, we can reproduce the key result from Sigdel et al. (2017) on how social norm strength affects cyclic dynamics (Figure 1 below). Yellow and orange regions exhibit dominance by one conservation opinion. Black regions exhibit bistability. The red and white regions exhibit an internal equilibrium, or persistent oscillations, respectively. The boundaries between these regions occur where the incentives to lead (Δ_L^1, δ_H^0) change sign. Further, the long run dynamics identified by Sigdel et al. (2017) in each region of the plot correspond exactly to what would be predicted by our models.

Our approach also allows us to interpret the results of Sigdel et al. (2017) in terms of the corner incentives. We find that social norms increase the incentive to be a follower of change, while simultaneously decreasing the incentive to be a leader of change. Once the incentive to be a leader of change becomes negative in one of the extreme environmental states, the possibility of cycles vanishes, and when norms are strong enough that the incentives to lead change become negative in both extreme environments, bistability results. Using our framework for eco-evolutionary games this surprising prediction about the relationship between social norms and cyclic dynamics can be plainly understood in terms of the key δ and Δ parameters we highlight, using Figure 2 (see supplementary information section S9). This discussion illustrates how our framework does not preclude the rich nuance found in cutting-edge environmental feedback research. In fact, our framework helps to understand why and how these findings arise.

Likewise, Sigdel et al. (2019) consider three different models of environmental dynamics, in linked social-ecological systems. The first model they consider is a special case of the renewable resource model that we analyze. They show that in some parameter regions, all three models of environmental dynamics produce similar outcomes. Similarly, we have shown that both renewing and decaying environmental dynamics produce the same outcomes. Taken together, these findings suggest that the findings of our paper are likely robust to the exact formulation of environmental dynamics (at least under some parameter regimes), and thus have broad applicability.

2. At the same time, the kind of dynamics that are possible to capture with this framework are standard and well-known in the associated literature, despite the fact that the authors present some of those findings as significant and new. For instance, I think few researchers in the field will be surprised to find that the dynamics of strategies and the environment depend on individual incentives to lead or follow behaviour change and on the relative speed of environmental versus strategic change, especially since the latter finding is already baked into the model assumptions.

Reply: We agree that it is intuitive that the incentives to lead and follow behavioral changes are critical to the dynamics of an eco-evolutionary system. But how exactly these incentives translate into dynamical outcomes is not so intuitive. For example, positive incentives to lead and follow environmental movements and gold rushes might naturally be expected to produce cyclical dynamics (Figure 1) – and yet our work shows these conditions are not necessary or sufficient for cycles.

Figure 1: The vertical axis is the strength of social norms, ξ , and the horizontal axis is the conservation value of forest, c . Yellow and orange regions exhibit dominance by one conservation opinion. Black regions exhibit bistability. The red and white regions exhibit an internal equilibrium, or persistent oscillations, respectively. In subfigure (a), $h/R = 1/2$ and in subfigure (b) $h/R = 2$.

More generally, we are unaware of any previous study that systematically delineates what combinations of incentives and time-scales leads to different dynamical regimes in eco-evolutionary games. And this is the primary value of our study.

Finally, some prominent prior work has shown the timescale of environmental change does not change the qualitative outcome of the system (Weitz et al., 2016). Whereas we find that the opposite is true, in general. We have explained this critical discrepancy to prior work by adding new analyses to the revised manuscript. The revised manuscript explains when and why timescale differences between the environmental and strategy dynamics will be important for dynamics outcomes. And so our work patently extends, and contextualizes, prior studies on eco-evolutionary games.

Also, it is not surprising that environmental feedbacks cause the system to exhibit richer dynamics, because the dimensionality of the system increases significantly when environmental feedbacks are added (and this has also already been reported in the literature).

Reply: We certainly agree – including environmental feedbacks is expected to lead to richer dynamics. What our work provides, however, is an exact description of the possible dynamical regimes that can be achieved in linear eco-evolutionary games, as well as extensions to non-linear cases. In the revised manuscript we have included a discussion of the role of dimensionality on the breadth of possible system dynamics.

Similarly, the cyclic dynamics report in Figure 1 are well known, even if they are not always described using the re-parameterization the authors introduce.

Reply: We agree. We do not claim that the cyclic dynamics in Figure 1 are novel in and of themselves. Rather, the re-scaling of the environment allows us to map models with disparate environmental dynamics to a common framework. And using this common framework we can derive exact conditions for when these (well known) cycles will, or will not, arise.

Overall, the authors have only provided evidence that the framework can represent the simpler

types of systems with environmental feedbacks, which limits use of the framework. Also, the predicted dynamics are not novel or surprising and have been exhibited by many other models in the literature—sometimes under other rubrics like coupled human-and-natural systems, but with the same dynamics nonetheless.

Reply: We agree that many particular models in socio-ecological systems theory have been analyzed and exhibit, say, cyclic or bistable dynamics; we do not claim these findings as novel. Indeed, we cite and discuss many of these papers in our manuscript, and we would be happy to include other relevant examples the referee might suggest.

Nonetheless, we are not aware of any other study that provides a synthetic framework for formulating and analyzing eco-evolutionary games – which is the primary value of our study.

References

- Rand, D. G., Tomlin, D., Bear, A., Ludvig, E. A., and Cohen, J. D. (2017). Cyclical population dynamics of automatic versus controlled processing: An evolutionary pendulum. *Psychological review*, 124(5):626.
- Sigdel, R., Anand, M., and Bauch, C. T. (2019). Convergence of socio-ecological dynamics in disparate ecological systems under strong coupling to human social systems. *Theoretical Ecology*, 12(3):285–296.
- Sigdel, R. P., Anand, M., and Bauch, C. T. (2017). Competition between injunctive social norms and conservation priorities gives rise to complex dynamics in a model of forest growth and opinion dynamics. *Journal of theoretical biology*, 432:132–140.
- Weitz, J. S., Eksin, C., Paarporn, K., Brown, S. P., and Ratcliff, W. C. (2016). An oscillating tragedy of the commons in replicator dynamics with game-environment feedback. *Proceedings of the National Academy of Sciences*, 113(47):E7518–E7525.

Reviewers' Comments:

Reviewer #1:

Remarks to the Author:

The authors have taken all my suggestions into account. In particular, they now explain more clearly how their framework relates to the earlier model of Weitz and colleagues, and how that earlier model can be represented as a special case of the current framework.

The authors have obviously put quite some effort into this revision, and I appreciate the changes they have made.

I think the revised paper satisfies all criteria for publication in Nature Communications.

There are only two minor comments:

(-) There are still numerous instances in both the main text and the SI in which the authors refer to "equilibria" or "equilibrium". Please refer to "fixed points" instead.

(-) There are still a few typos in both the main text and SI. While I assume that the typos in the main text will be taken care of during the editing process, please proofread the SI once more (especially the new Sections S8 and S9). A few examples:

Line 1163: "These equilibria will always be unstable, either unstable nodes or saddle equilibria depending on whether incentive..." — I assume the correct sentence is "These fixed points will either be unstable or saddle equilibria depending on whether the incentive"

Line 1183: "the stability criteria does not depend" -> "the stability criteria do not depend"

Line 1200: "Analysis can of this system can be simplified" -> "Analysis of this system can be simplified"

After Line 1212: "The criteria for an unstable interior equilibrium in this model are different that those" -> "... are different from those"

Reviewer #2:

Remarks to the Author:

The authors' first round of revisions have addressed most of my previous concerns about novelty and expressing limitations of the approach. I am still not sure how easy it is to generalize about social-ecological systems, compared to less messy systems in physics for instance. However, that criticism can be levelled against any such framework in social-ecological systems, and even a limited ability to do so could be useful. The revised version better shows that their framework can capture a diverse set of model systems and provide a unifying language for the dynamics, especially with the expanded analysis of the revised manuscript.

I have a few remaining concerns that could be addressed through some minor re-writing:

1. Line 484-504 contains a nice discussion of heterogeneity but I feel this material is not adequately connected to the rest of the paper. The authors essentially say that some systems are just too complicated to be captured by the framework... so the reader is left wondering what to do with the authors' framework in these cases. How easy is it to determine whether a given system can be expressed through their framework? Can higher-dimensional systems be reduced to the four delta parameters as easily as for the 2D case? It would be helpful if the authors could provide some guidance/speculation as to how others in future work might approach the problem of expressing more complex systems through this framework, or how easily it can be extended to more complicated cases.

2. Following on comment #1, the abstract should convey some of the potential limitations of the framework, or at least mention that it has limits. At present, the abstract sounds like this framework will apply to any eco-evolutionary system.

3. The background literature is a little too focused on evolution-derived work, despite the fact that the mathematical equations are often identical across systems in ecology, population dynamics, environment, and sociology. Below are a few more papers to cite that exhibit dynamics similar to the ones explored by the authors, and/or exhibit the breadth of possible social-ecological model systems it might be possible to capture with their framework. The range of applications of this modelling approach is vast, as the authors are already aware.

- Lacitignola, Deborah, et al. "Modelling socio-ecological tourism-based systems for sustainability." *Ecological Modelling* 206.1-2 (2007): 191-204. (demonstrates breadth of applications)

- Innes, Clinton, Madhur Anand, and Chris T. Bauch. "The impact of human-environment interactions on the stability of forest-grassland mosaic ecosystems." *Scientific reports* 3 (2013): 2689. (early example of replicator approach to social-ecological systems)

- Richter, Andries, and Vasilis Dakos. "Profit fluctuations signal eroding resilience of natural resources." *Ecological Economics* 117 (2015): 12-21. (economic aspects of SES)

- Lee, Joung Hun, et al. "Coupled social and ecological dynamics of herders in Mongolian rangelands." *Ecological Economics* 114 (2015): 208-217. (demonstrates breadth of applications)

Response to Reviewers

for: *Evolutionary games with environmental feedbacks*
Andrew R. Tilman, Joshua B. Plotkin, and Erol Akçay

Reviewer 1

The authors have taken all my suggestions into account. In particular, they now explain more clearly how their framework relates to the earlier model of Weitz and colleagues, and how that earlier model can be represented as a special case of the current framework.

The authors have obviously put quite some effort into this revision, and I appreciate the changes they have made.

I think the revised paper satisfies all criteria for publication in Nature Communications. There are only two minor comments:

- There are still numerous instances in both the main text and the SI in which the authors refer to “equilibria” or “equilibrium”. Please refer to “fixed points” instead.

Reply: We hesitate to replace each instance of ‘equilibrium’ with ‘fixed point.’ While ‘fixed point’ is precise and technically correct, it is also a term that will be less familiar to many readers. To resolve this issue we now explicitly define our equilibrium notion, on lines 185-186.

- There are still a few typos in both the main text and SI. While I assume that the typos in the main text will be taken care of during the editing process, please proofread the SI once more (especially the new Sections S8 and S9). A few examples:

Line 1163: “These equilibria will always be unstable, either unstable nodes or saddle equilibria depending on whether incentive...” — I assume the correct sentence is “These fixed points will either be unstable or saddle equilibria depending on whether the incentive”

Line 1183: “the stability criteria does not depend” → “the stability criteria do not depend”

Line 1200: “Analysis can of this system can be simplified” → “Analysis of this system can be simplified”

After Line 1212: “The criteria for an unstable interior equilibrium in this model are different that those” → “...are different from those”

Reply: We thank the reviewer for noticing these errors. We have corrected them and carefully checked for further errors.

Reviewer 2

The authors’ first round of revisions have addressed most of my previous concerns about novelty and expressing limitations of the approach. I am still not sure how easy it is to generalize about social-ecological systems, compared to less messy systems in physics for instance. However, that criticism can be levelled against any such framework in social-ecological systems, and even a limited ability to do so could be useful. The revised version better shows that their framework can capture a diverse set of model systems and provide a unifying language for the dynamics, especially with the expanded analysis of the revised manuscript.

I have a few remaining concerns that could be addressed through some minor re-writing:

1. Line 484-504 contains a nice discussion of heterogeneity but I feel this material is not adequately connected to the rest of the paper. The authors essentially say that some systems are just too complicated to be captured by the framework... so the reader is left wondering what to do with the authors' framework in these cases. How easy is it to determine whether a given system can be expressed through their framework? Can higher-dimensional systems be reduced to the four delta parameters as easily as for the 2D case? It would be helpful if the authors could provide some guidance/speculation as to how others in future work might approach the problem of expressing more complex systems through this framework, or how easily it can be extended to more complicated cases.

Reply: The approach taken in our paper of writing payoffs to a linear game in terms of a re-scaled environmental metric will not apply to all systems. Of particular interest is how this approach could be used in games with more than two strategies. In this case, there are more strategy combinations between which individuals could lead and follow strategy change. This will necessarily increase the dimensionality of the parameter space beyond the four delta parameters that describe most of the dynamics of the linear two-strategy system. We now include in the revised discussion a description of our intuition that incentives to lead and follow can still provide significant insight into eco-evolutionary games with more than two strategies.

2. Following on comment #1, the abstract should convey some of the potential limitations of the framework, or at least mention that it has limits. At present, the abstract sounds like this framework will apply to any eco-evolutionary system.

Reply: Given the length limit, in our revised abstract we convey as best we can an overview of the questions that we seek to address in the manuscript. Our analysis, of course, has limits, and we state these limits clearly and explicitly in the text (that our in-depth analysis covers two-strategy linear eco-evolutionary games, for example.)

3. The background literature is a little too focused on evolution-derived work, despite the fact that the mathematical equations are often identical across systems in ecology, population dynamics, environment, and sociology. Below are a few more papers to cite that exhibit dynamics similar to the ones explored by the authors, and/or exhibit the breadth of possible social-ecological model systems it might be possible to capture with their framework. The range of applications of this modelling approach is vast, as the authors are already aware.

Reply: We thank the reviewer for bringing these relevant references to our attention. We have incorporated them into the introduction of the revised manuscript.

- Licitignola, Deborah, et al. "Modelling socio-ecological tourism-based systems for sustainability." *Ecological Modelling* 206.1-2 (2007): 191-204. (demonstrates breadth of applications)
- Innes, Clinton, Madhur Anand, and Chris T. Bauch. "The impact of human-environment interactions on the stability of forest-grassland mosaic ecosystems." *Scientific reports* 3 (2013): 2689. (early example of replicator approach to social-ecological systems)
- Richter, Andries, and Vasilis Dakos. "Profit fluctuations signal eroding resilience of natural resources." *Ecological Economics* 117 (2015): 12-21. (economic aspects of SES)

- Lee, Joung Hun, et al. “Coupled social and ecological dynamics of herders in Mongolian rangelands.” *Ecological Economics* 114 (2015): 208-217. (demonstrates breadth of applications)